# LaMAGIC2: Advanced Circuit Formulations for Language Model-Based Analog Topology Generation

**Chen-Chia Chang** [1]  **Wan-Hsuan Lin** [2]  **Yikang Shen** [3]  **Yiran Chen** [1]  **Xin Zhang** [3][4]

## Abstract

Automation of analog topology design is crucial due to customized requirements of modern applications with heavily manual engineering efforts. The state-of-the-art work applies a sequence-to-sequence approach and supervised finetuning on language models to generate topologies given user specifications. However, its circuit formulation is inefficient due to $O(|V|^2)$ token length and suffers from low precision sensitivity to numeric inputs. In this work, we introduce LaMAGIC2, a succinct float-input canonical formulation with identifier (SFCI) for language model-based analog topology generation. SFCI addresses these challenges by improving component-type recognition through identifier-based representations, reducing token length complexity to $O(|V|)$, and enhancing numeric precision sensitivity for better performance under tight tolerances. Our experiments demonstrate that LaMAGIC2 achieves 34% higher success rates under a tight tolerance of 0.01 and 10X lower MSEs compared to a prior method. LaMAGIC2 also exhibits better transferability for circuits with more vertices with up to 58.5% improvement. These advancements establish LaMAGIC2 as a robust framework for analog topology generation. Code available at https://github.com/turtleben/LaMAGIC.

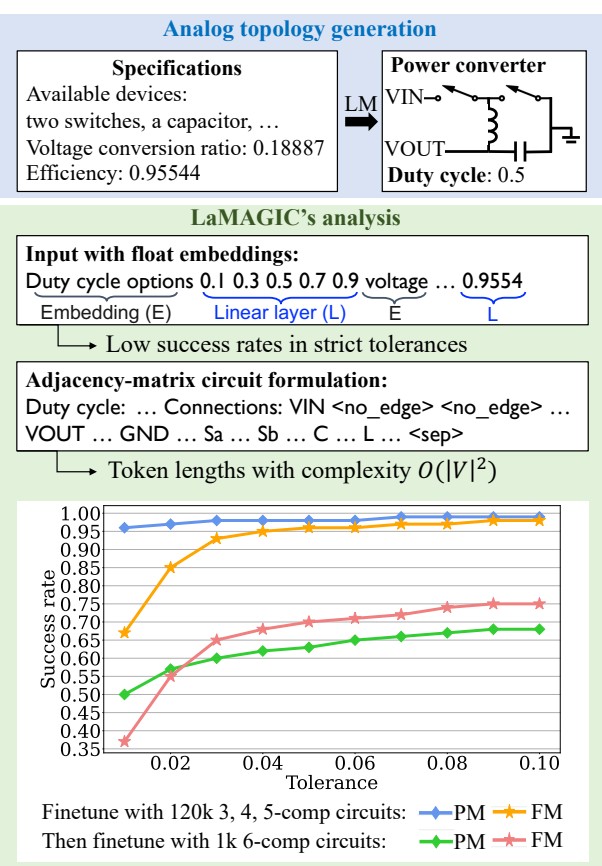

*Figure 1.* Analog topology generation and the analysis of a state-of-the-art work LaMAGIC (Chang et al., 2024).

## 1. Introduction

With the rise of diverse electronic systems, the need for different analog circuit functionalities increases. Thus, the demand for analog topology customization has largely increased. For example, the power converter application with different power supply requirements, e.g., voltage conversion ratio and power efficiency, often requires varied topologies. However, traditional topology design relies heavily on manual processes, which demand large engineering efforts and prolong the time-to-market period for new designs. Thus, automating analog topology design has become crucial to accelerate circuit development.

Early efforts (Fan et al., 2021; Zhao & Zhang, 2022; Lu et al., 2023) have focused on search-based approaches, which explore vast design spaces guided by simulation rewards. However, they are inefficient and are not suitable to generate circuits for diverse specifications. The work (Fan et al.,

---

[1]Duke University [2]University of California, Los Angeles [3]MIT-IBM Watson AI Lab [4]IBM T. J. Watson Research Center. Correspondence to: Chen-Chia Chang <chenchia.chang@duke.edu>, Xin Zhang <xzhang@us.ibm.com>.

*Proceedings of the 42nd International Conference on Machine Learning*, Vancouver, Canada. PMLR 267, 2025. Copyright 2025 by the author(s).

*Table 1.* Average token lengths of LM outputs from: SFCI in LaMAGIC2 and FM in LaMAGIC (Chang et al., 2024).

| Average token length | 3, 4, 5-comp circuits | 6-comp circuits |
| --- | --- | --- |
| LaMAGIC2-SFCI | 36.40 | 41.41 |
| LaMAGIC-FM | 84.79 | 105.00 |

2021) designs a upper-confidence-bound-tree-based reinforcement learning (RL) method for power converter design, which requires hundreds of simulation queries to generate a new topology. The other work (Zhao & Zhang, 2022) also develops an RL algorithm for operational amplifiers, which takes 73 simulation iterations for 4 hours per specification. Finally, the work (Lu et al., 2023) uses Bayesian optimization for topology search. However, 50 iterations are needed for one specification.

Recent advances in generative modeling and large language models (LLMs) (Radford et al., 2019; Raffel et al., 2020; Chung et al., 2022; Nijkamp et al., 2023) offer an alternative paradigm. Instead of exhaustively exploring the design space, generative models can learn direct mappings from performance requirements to circuit topologies. One such approach is LaMAGIC (Chang et al., 2024), which frames circuit generation as a sequence-to-sequence problem for transformer-based autoregressive language models. LaMAGIC proposes several circuit formulations to perform supervised finetuning (SFT) (Chung et al., 2022; Taori et al., 2023). The trained models can generate a topology via one model inference, as shown in Figure 1, significantly accelerating the generation process compared to prior search methods.

While LaMAGIC demonstrates great potential for analog topology generation, its circuit formulations pose notable limitations, as shown in Figure 1. First, its formulation reduces the model's sensitivity to low precision difference of numeric inputs, resulting in low success rates under strict tolerance evaluations. Second, the token length of its formulations scales quadratically with the number of nodes $O(|V|^2)$. The long output sequences pose significant challenges to accuracy due to error accumulation and context window limitation for transformer-based topology generation. As generation progresses autoregressively, early errors propagate, compounding inaccuracies and reducing the reliability of generated circuit structures. Additionally, when output exceeds the model's fixed context window, critical early design elements may fall out of scope, leading to inconsistencies or infeasible circuit configurations. These limitations are particularly problematic in topology generation, where precise structural relationships must be maintained.

In this work, we propose LaMAGIC2, an advanced circuit formulation designed for LM-based topology generation. LaMAGIC2 improves low-precision sensitivity to numeric

inputs by shrinking the LM inputs. In addition, we address the long token length problem by reducing the complexity from $O(|V|^2)$ to $O(|V|)$ while incorporating component-type tokens. This reduction is particularly impactful for real-world circuits, whose node numbers are large. Thus, LaMAGIC2 can shorten the token length, as shown in Table 1, to mitigate the long sequence challenge faced by transformer models.

Our contributions are summarized as follows.

- We propose succinct float-input canonical formulation with identifier (SFCI): A sparse and canonical representation that uses identifiers to enhance learning of component types while maintaining linear token growth relative to circuit size.
- LaMAGIC2 achieves a 122% and 34% higher success rate under a low tolerance 0.01 compared to an RL search method (Fan et al., 2021) and LaMAGIC, respectively.
- Models trained with SFCI demonstrate superior performance in limited data scenarios for more complex circuits, achieving the highest success rates to outperform other formulations.

Furthermore, our step-by-step analysis of formulations provides valuable insights into graph generation with transformer models, advancing the field of topology generation and beyond.

## 2. Preliminaries

### 2.1. Analog Topology Design

In this work, following LaMAGIC (Chang et al., 2024), we target on power converter topology generation, which aims to produce customized power converters that achieve specific design specifications, i.e., the voltage conversion ratio and the power conversion efficiency. The voltage conversion ratio is the ratio of output to input voltages. The power conversion efficiency is the ratio of output power to input power. Another design choice for power converter is the duty cycle with a range of 0 to 1, which controls the ON times of switches, influencing the output voltage and efficiency. In our framework, we consider five duty cycle options: {0.1, 0.3, 0.5, 0.7, 0.9}.

We view the circuit topology as a hypergraph $G$ with vertices $V$ and hyperedges $E$. For each topology, the vertices $V$ contain three terminal ports and several analog components. The three terminal ports are: a voltage input $VIN$, a voltage output $VOUT$, and a ground $GND$, each with an edge to connect to other vertices. User can pick four types of components: capacitors $C$, inductors $L$, phase-I switches $Sa$, and phase-II switches $Sb$, each with two edges. Each hyperedge $e \in E$ is a set of vertices that represents connections between devices and ports. An example of the

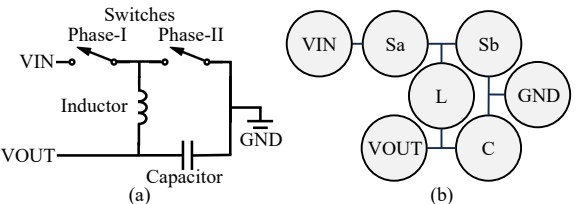

*Figure 2.* (a) An example power converter circuit and (b) its corresponding hypergraph representation. Note that here we use the same example in LaMAGIC (Chang et al., 2024).

power converter and its hypergraph representation is shown in Figure 2.

We utilize the same dataset in LaMAGIC (Chang et al., 2024). It contains 3, 4, 5-component circuits with 120k data points for training and 12k for evaluation. To assess the transferability of models to more complex circuits, the dataset has 76k unique 6-component circuits and split 9k data points for evaluation. In our experiments, we randomly select subsets of 500, 1k, and 2k 6-component circuits to fine-tune models initially trained on the 120k 3, 4, 5-component circuits.

Generating complex circuits is a critical challenge for human designers. Also, LaMAGIC does not perform well for transferability to larger circuits. Thus, our work focuses on improving this capability by leveraging input specifications that have performance targets and component requirements.

Based on these considerations, we define our problem:
*Problem.* Given vertices $V$, a voltage conversion ratio $r$, and an efficiency $\eta$, the model generates connections $E$ and determine the duty cycle $s \in \{0.1, 0.3, 0.5, 0.7, 0.9\}$ to build a circuit such that its simulation performance satisfies both $r$ and $\eta$.

## 2.2. Autoregressive Language Modeling

Autoregressive language models (LMs) (Radford et al., 2019; Raffel et al., 2020; Chung et al., 2022; Nijkamp et al., 2023; Zhao et al., 2023) are widely used in sequence modeling tasks. They predict the next token in a sequence based on preceding tokens, optimizing the autoregressive loss: $\ell = -\sum_{i=1}^{n} \log P(x_i | x_1, x_2, ..., x_{i-1})$. Building on LaMAGIC (Chang et al., 2024), this work also employs autoregressive LMs to address analog topology generation. These models are well-suited for this task as they can sequentially generate the next component and connection in a circuit based on the evolving subgraph.

## 2.3. Related Works on Topology Generation

Search-based methods (Fan et al., 2021; Zhao & Zhang, 2022; Lu et al., 2023) explore design spaces using simulation feedback but require significant computational re-

sources. AnalogCoder (Lai et al., 2024) leverages prompt engineering in general-purpose LLMs to iteratively refine circuit designs based on simulation feedback. However, its methodology is fundamentally different from our work, as it does not involve SFT and tailor formulations for direct specification-to-topology mapping. Our work builds upon LaMAGIC (Chang et al., 2024), which introduced custom circuit formulations to enable precise specification-to-topology generation. Given the drawbacks of LaMAGIC's formulations provided in our analysis in Section 3, we propose novel formulations to address these limitations.

## 3. Analysis of Formulations in LaMAGIC

LaMAGIC (Chang et al., 2024) introduces three formulations for circuit generation: (1) canonical formulation (CF), (2) pure-text adjacency-matrix formulation (PM), and (3) float-input adjacency-matrix formulation (FM), as shown in Figure 3. In this section, we analyze advantages and disadvantages of each formulation.

### 3.1. Canonical Formulation (CF)

CF encodes circuit information with a canonical representation with output edges sorted based on a predefined input vertex order.

**Advantage:**
(1) Compact circuit encoding: Since each edge is represented by a set of nodes, the length of CF is $O(|V|)$.

**Disadvantage:**
(1) Limited component type awareness: LaMAGIC's tokenizer for CF encodes node-specific tokens (e.g., `Sa0`, `Sa1`) into separate single embeddings (e.g., `<Sa0>`, `<Sa1>`). This method make models difficult to learn relations between different component types because models cannot easily recognize the component type of each node, e.g., `Sa0` and `Sa1` represent the same node type `Sa`. This drawback can be the reason of the bad generalizability of CF.

(2) Numeric tokenization: When numeric values (e.g., `0.95544`) are fed into the tokenizer, they are split into multiple subword tokens (e.g., "0", ".", "9", "5", ...), diluting their semantic meaning and making it harder for the model to learn numeric relationships.

### 3.2. Pure-Text Adjacency-Matrix Formulation (PM)

PM represents circuit connections as an adjacency matrix for hypergraph, where rows and columns are indexed based on the vertex order given in the input.

**Advantage:**
(1) Component type recognition: Because the matrix indexes are inherently ordered by the input vertex sequence, PM eliminates the need to assign identifiers to distinguish

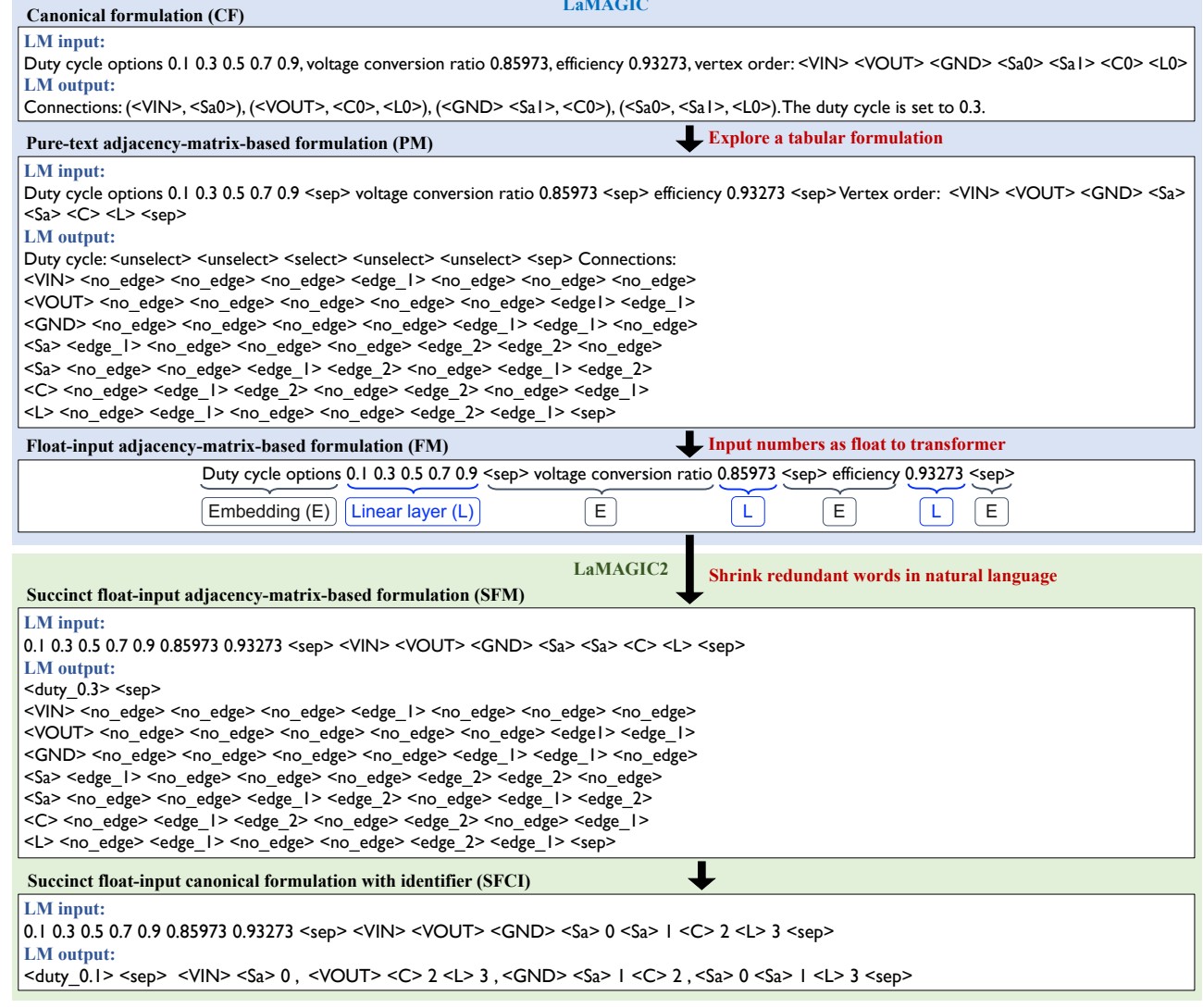

Figure 3. The circuit formulations proposed by LaMAGIC (Chang et al., 2024) and our work LaMAGIC2. Tokens enclosed within < and > denote those added to the tokenizer's dictionary, enabling distinct embeddings for each token.

nodes of the same component type. For example, instead of using `<Sa0>` and `<Sa1>`, PM can simply use `<Sa>` to represent multiple nodes of the same type. Therefore, PM enhances the model's ability to recognize relationships between nodes of the same type, improving its understanding of circuit structures and generalizability.

(2) Graph difference detection: This tabular formulation excels at graph difference detection during training because errors are localized to specific matrix entries. For example, a missing or incorrect edge affects only the corresponding matrix entry, making it easier for the model to identify and correct isolated errors.

**Disadvantage:**
(1) Long token length $O(|V|^2)$: Longer sequences increase

attention complexity, which leads to ineffectiveness in transformer models.

(2) Numeric tokenization, the same as in CF.

### 3.3. Float-Input Adjacency-Matrix Formulation (FM)

FM addresses PM's numeric tokenization issues by incorporating a shared linear layer to encode numeric inputs directly into the transformer's embedding space.

**Advantage:**
(1) Float-input numerical encoding: This method strengthens the model's capacity to learn the relationship between input specifications and circuit structure. As a result, FM demonstrates strong transferability, particularly when generating circuits with six components.

**Disadvantage:**
(1) Low sensitivity to numeric precision: FM exhibits low success rates for 345-component circuits under strict tolerance conditions, achieving only 0.67 compared to PM's 0.93 at a tolerance level of 0.01, as illustrated in Figure 1. This limitation could arise from FM's input formulation, which includes redundant natural language descriptions (e.g., duty cycle and voltage) that dilute the model's focus on numeric inputs. This reduces the model ability to capture fine-grained differences between numeric inputs.

### 3.4. Summary of Insights and Challenges

The formulations in LaMAGIC (Chang et al., 2024) highlight key insights: (1) the component-type token representation is crucial for learning circuit structures, and (2) integrating numerical inputs with a shared linear layer enhances generalization for complex circuits. However, these approaches face challenges: (1) matrix-based formulations suffer from inefficient token lengths $O(|V|^2)$, and (2) the float-input setting has low sensitivity to numeric input precision.

To address these issues, our proposed formulations combine the strengths of FM and PM by enhancing numeric attention, reducing token length to scale linearly with $|V|$, and improving component type representation for better graph structure learning. These improvements enable more robust circuit generation.

## 4. Our Proposed Formulations

This work introduces two novel circuit representations: (1) succinct float-input adjacency-matrix-based formulation and (2) succinct float-input canonical formulation with identifier, as illustrated in Figure 3.

### 4.1. Succinct Float-Input Adjacency-Matrix Formulation (SFM)

SFM improves upon the matrix-based and float-input settings of FM, by reducing unnecessary details in the representation to better handle numeric inputs. Its key components are as follows:

**1. Succinct input:** Because matrix formulation is deviated from natural language, the natural language descriptions, e.g., "Duty cycle" and "voltage", cannot provide effective signal for topology generation. SFM removes these descriptions from the input, creating a representation that focuses on the essential numerical inputs. This representation reduces attention dilution, enabling the model to concentrate on learning a direct mapping between numeric inputs and the adjacency matrix. As a result, SFM enhances the model's precision sensitivity and the performance under strict tolerance conditions.

**2. Simplified representation for duty cycle selection:** SFM simplifies duty cycle selection by replacing the five-token representation <unselect> <unselect> <select> <unselect> <unselect> in FM and PM with a single token <duty_0.3>.This change better aligns with the classification nature of duty cycle selection, where the task involves choosing one option from five predefined choices <duty_0.1>, <duty_0.3>, <duty_0.5>, <duty_0.7>, or <duty_0.9>. Since autoregressive LMs inherently operate at the token level, treating this decision as a single-token classification task simplifies learning and prediction. We add these five tokens into the dictionary of the tokenizer to let each one represented by a distinct embedding, ensuring clarity in the model's understanding. This formulation not only reduces sequence length but also leverages the natural behavior of LMs to enhance generation effectiveness.

### 4.2. Succinct Float-Input Canonical Formulation with Identifier (SFCI)

As SFM shrinks the input and output by removing natural language descriptions and simplifying duty cycle representation, matrix-related formulations still face challenges due to their long token length, $O(|V|^2)$. Additionally, real circuits are often sparse, meaning that most entries in the adjacency matrix are <no_edge> tokens. This results in an inefficient representation, as the model expends capacity processing numerous tokens that carry minimal informational value. As $|V|$ increases, matrix-related formulations could struggle to handle large circuit generation effectively.

We observe that CF performs well for circuits with 3, 4, 5 components but lacks component-type tokens, which are important for distinguishing node types. We believe this limitation contributes to its lower performance in generating circuits with six components. In contrast, the matrix-based formulations (PM, FM), who has better transferability, all utilize node-type tokens (e.g., <Sa> and <Sb> as single tokens). To address these limitations, we propose the succinct float-input canonical formulation with identifier (SFCI).

SFCI adopts the sparse graph representation of CF, where edges are represented as sets of nodes, and introduces the following design choices:

**1. Identifiers for device nodes:** SFCI separates each node token like Sa0 into two tokens: the device type (<Sa>) and an identifier (0). Identifiers are used only for device tokens, as circuits often contain similar devices. The identifiers are incremented from 0 to $|V-3|$, where $|V-3|$ is the number of devices in the circuit. Three port nodes (<VIN>, <VOUT>, <GND>) do not use identifiers, as each port occurs only once in a circuit. This approach allows SFCI to improve the component type recognition.

**2. Simplified output format:** SFCI removes bracket tokens

---

**SFCI without Component-Type Tokens in Output (SFCI-NCT)**

> **LM input:**
> (Same as SFCI)
> **LM output:**
> <duty_0.1> <sep> <VIN> 0 , <VOUT> 2 3 ,<GND> 1 2 ,0 1 3 <sep>

**SFCI without Duty-Cycle Prefix (SFCI-NDP)**

> **LM input:**
> 0.85973 0.93273 <sep> <VIN> <VOUT> <GND> <Sa> 0 <Sa> 1 <C>
> 2 <L> 3 <sep>
> **LM output:**
> (Same as SFCI)

*Figure 4.* Two variants of SFCI: (1) without component-type token in output (SFCI-NCT) and (2) without a common feature duty-cycle prefix (SFCI-NDP).

( ) and uses commas , to separate the node set for each edge. For example, an edge is represented as <VIN> <Sa> 0 rather than a verbose bracketed format. This reduces the token count to simplify the learning process.

**3. Component-type tokens in the output:** If we would like to ultimately shrink SFCI, we can eliminate the component-type tokens in the output to be SFCI-NCT in Figure 4. Although this method can reduce output lengths by nearly half, we retain component-type tokens (e.g., <Sa>, <Sb>, <C>, <L>) in the output of SFCI. Because SFCI's outputs are self-contained without relying on input-output fusion to understand the circuit, these tokens can provide explicit patterns that map identifiers to their component types in the circuit to facilitate model learning. The effectiveness of this design choice is validated in Section 6.2.

**Token length complexity of SFCI:** SFCI reduces the token length of the output to $O(|V|)$. Each hyperedge $e_i$ has $k_i$ vertices. Thus, the total token length, representing the sum of vertex incidences across all $e$, is $\sum_{i=0}^{|E|} k_i$. For our power converter devices, each vertex appears in at most two hyperedges: $d(v) < 2$. Thus, the total number of vertex incidences, which equals to the number of edges of all nodes, is $\sum_{i=0}^{|E|} k_i = \sum_{v \in V} d(v) \leq 2|V|$. This upper bound implies the total token length is at most $2|V|$, which simplifies to $O(|V|)$.

This result indicates that the token length complexity of SFCI is independent of the number of edges $|E|$. The $O(|V|)$ token length of SFCI demonstrates its compactness compared to $O(|V|^2)$ lengths of matrix formulations. We compute the average token length of outputs from SFCI and FM in Table 1. Token length of SFCI only increases five when the dataset grows to 6-component circuits, while token length of FM grows quadratically from 73 to 105. By combining the strengths of CF's sparse representation with the use of component-type tokens and identifiers, SFCI balances compactness and expressiveness, enabling robust generation of large circuits while addressing the limitations of canonical and matrix-based formulations.

We observe that all formulations include a duty-cycle prefix ranging from 0.1 to 0.9, a common feature across all input-output pairs. To make formulations succinct, one might consider removing this redundancy, as shown in SFCI-NDP (Figure 4). However, this duty-cycle prefix has the following two benefits. First, the duty-cycle prefix serves as implicit regularization, helping the model stabilize its representations. Second, it provides guidance for decision-making by acting as a valid set of duty cycle options the model can reference throughout training. Without this prefix, the model must infer these duty-cycle values purely from other inputs. The effectiveness of the prefix is established in Section 6.1.

## 5. Experimental Results

### 5.1. Experiment Setup

**Baselines.** To evaluate the performance of our proposed formulations SFM and SFCI, we compare them with three baseline formulations introduced in LaMAGIC (Chang et al., 2024): canonical formulation (CF), pure-text adjacency-matrix-based formulation (PM), and float-input adjacency-matrix-based formulation (FM), as introduced in Section 3.

In addition to SFT methods that directly generate the circuit in one-shot, we compare with a prior work (Fan et al., 2021) that uses an RL search algorithm for power converter generation. They need to run simulator each query to give feedback to the RL engine. We set the total query budget to 100 per input specification. We also compare with an advanced reasoning model o1 (Jaech et al., 2024) using a few-shot setup with 100 example circuits as context.

**Training details.** We follow LaMAGIC (Chang et al., 2024), adopting the Flan-T5-base encoder-decoder transformer (Chung et al., 2022) initialized with pretrained weights. To handle numeric inputs, the standard word embedding layer is replaced by a shared linear projection. Training employs conditional generation to map input-output pairs, with random vertex order permutations applied as data augmentation (Chang et al., 2024). Customized tokens are added to tokenizer's dictionary to represent component types and duty cycle options. For SFM, we add <sep>, <duty_0.1>, <duty_0.3>, <duty_0.5>, <duty_0.7>, <duty_0.9>, VIN, VOUT, GND, Sa, Sb, C, L, <no_edge>, <edge_1>, <edge_2>, <both_edges>. For SFCI, we further add identifier tokens 0 to 12.

Flan-T5-base contains 248M parameters with 12 layers each in encoder and decoder, 64-dimensional key/value projections, 2048-dimensional feed-forward layers, and 12 attention heads. Training runs on one NVIDIA V100 GPU using AdamW with the following hyperparameter: learning rate $3 \times 10^{-4}$, cosine scheduler with 300 warm-up steps, batch size 128, L2 regularization $10^{-5}$, dropout 0.1, and epochs

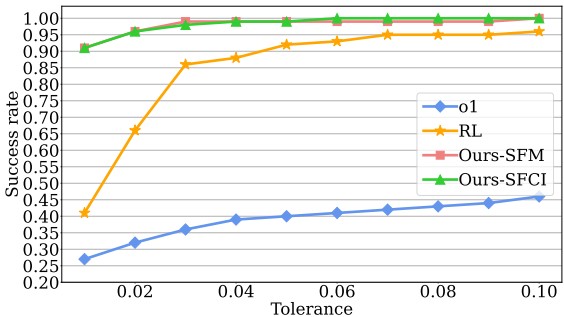

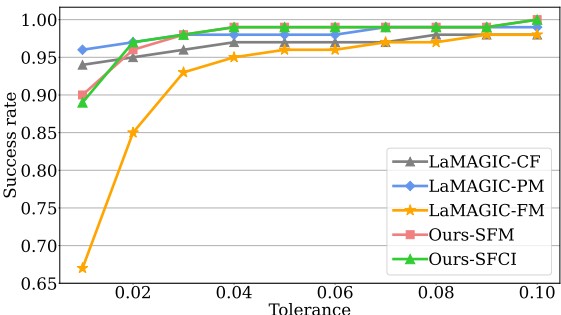

*Figure 5.* Success rates of an RL-search method (Fan et al., 2021) and models trained with our circuit formulations SFM and SFCI using 3, 4, 5-component circuits.

*Figure 6.* Success rates of models trained with different circuit formulations using 3, 4, 5-component circuits.

*Table 2.* MSEs of an RL-search method (Fan et al., 2021), o1 with few-shot prompting, and models trained with our circuit formulations SFM and SFCI using 3, 4, 5-component circuits.

| MSE | RL | o1 | SFM | SFCI |
|---------|-------|-------|--------|---------|
| Voltage | 0.313 | 0.602 | 0.0001 | 0.00008 |

*Table 3.* MSEs of models trained with different circuit formulations using 3, 4, 5-component circuits.

| MSE | Voltage | Efficiency |
|------|---------|------------|
| CF | 0.0160 | 0.0040 |
| PM | 0.0065 | 0.0023 |
| FM | 0.0061 | 0.0128 |
| SFM | 0.0013 | 0.0003 |
| SFCI | 0.0006 | 0.0002 |

120.

**Evaluation metrics.** Our primary metrics for evaluation are (1) the success rate of the generated circuits within varied tolerances $t$ ranging from 0.01 to 0.1 and (2) the mean squared error (MSE) between the input specifications and the simulated performance of the generated circuits. We run simulator NGSPICE (Nenzi P, 2011) on each generated circuit to get its actual voltage conversion ratio and efficiency for real-world applications.

The success rate is the proportion of generated circuits whose simulated voltage conversion ratio $v$ and efficiency $e$ fell within a tolerance $t$ of the target input specifications $v'$ and $e'$, i.e. $v$ and $e$ are both within the range $v' \pm t$ and $e' \pm t$. If a generated circuit is unsimulable (invalid), we mark it as an unsuccessful one when calculating success rates. MSEs for voltage conversion ratio and efficiency are computed separately. In addition, the invalid generated circuit is viewed as error 1 in MSE calculation.

### 5.2. Generation Results on 3, 4, 5-Component Circuit

**Comparison with RL-search method and o1.** We run RL-search method (Fan et al., 2021) for 5 days to complete the generation of 350 specifications from our testing set. Thus, we will compare this work (named RL) and o1 with our formulations under the same 350 specifications. Since RL only constrains voltage conversion ratio in topology generation, we evaluate the performance on success rates and MSE calculated only on voltage conversion ratios. As shown in Figure 5 and Table 2, our SFT methods under SFM and SFCI largely outperform RL (Fan et al., 2021), with success rates 0.41 (RL) and 0.91 (SFM and SFCI) on a tight

tolerance 0.01. In addition, o1 does not perform well with few-shot prompting, indicating the need of SFT for analog tolopogy generation.

**Comparison between different formulations.** We perform SFT under CF, PM, FM, SFM, and SFCI for 3,4,5-component circuits and evaluate them using the testing set containing 12k data points.

From LaMAGIC (Chang et al., 2024), FM does not performs well on 3,4,5-component circuits due to its float-input setting but it has better generalizability to more complex circuits. As shown in Figure 6, our SFM and SFCI successfully improve the success rates under tight tolerance ranges compared to FM (from rates 0.67 to 0.9 under a tolerance of 0.01). In addition, both SFM and SFCI become comparable to PM that uses the all-text-input setting. Specifically, our success rates are 0.99 when tolerances are larger than 0.04 and 1.00 when the tolerance is 0.1.

Under MSE in Table 3, SFM and SFCI significantly outperform all baseline formulations, especially for SFCI that shows 10X lower MSE compared to FM and PM. SFM's result shows that our succinct formulation benefits the float input setting to let models better learn the mapping between numerical numbers and edges. Additionally, comparison between SFCI and CF shows that component-type tokens can improve CF's sparse formulation in circuit generation.

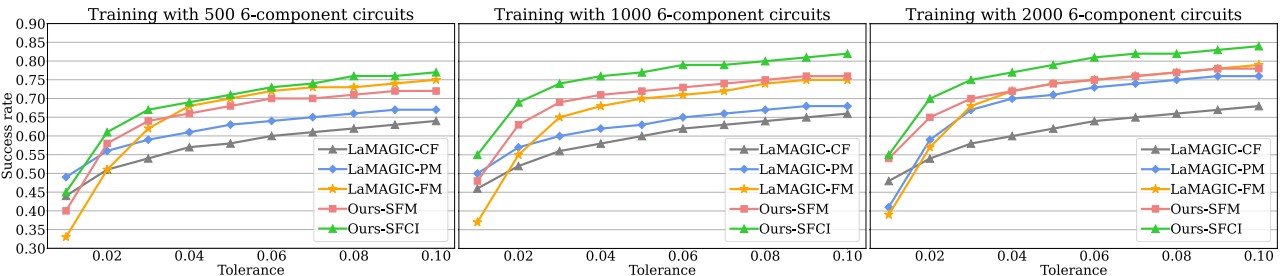

*Figure 7.* Success rates of models finetuned with different circuit formulations using 500, 1000, and 2000 6-component circuits.

*Table 4.* MSEs of voltage conversion ratio and efficiency evaluated on models finetuned with 500, 1k, and 2k 6-component circuits.

| MSE | 500 | | 1000 | | 2000 | |
|---|---|---|---|---|---|---|
| | Voltage | Eff | Voltage | Eff | Voltage | Eff |
| CF | 0.1843 | 0.1970 | 0.1684 | 0.1844 | 0.1459 | 0.1661 |
| PM | 0.1661 | 0.1705 | 0.1494 | 0.1565 | 0.1334 | 0.1315 |
| FM | 0.1324 | 0.1156 | 0.1341 | 0.1325 | 0.1014 | 0.0865 |
| SFM | 0.1570 | 0.1543 | 0.1188 | 0.1109 | 0.0941 | 0.1009 |
| SFCI | 0.1102 | 0.0899 | 0.0475 | 0.0719 | 0.0580 | 0.0418 |

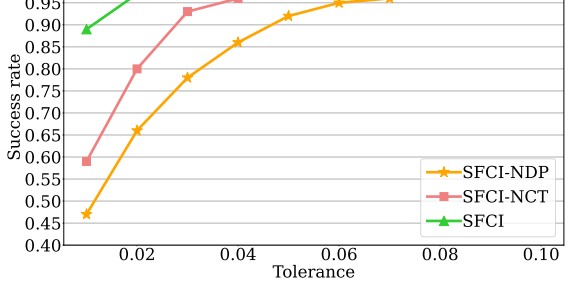

*Figure 8.* Success rates of models trained with SFCI and its two variants using 3, 4, 5-component circuits: (1) SFCI-NDP and (2) SFCI-NCT, as in Figure 4.

## 5.3. Transferability Evaluation on 6-Component Circuit

In real world scenario, large amount of data for circuits with a large node number could be hard to collect. A six-component circuit requires 30 seconds of simulation to obtain the voltage and efficiency. Similar to LaMAGIC, we extend models trained with 3, 4, 5-components circuits to be finetuned with only 500, 1k, and 2k 6-component circuits randomly selected from our dataset. Then, we evaluate each model on the testing set (9k data points) and run simulation on each generated circuit to get its real performance.

The results are shown in Figure 7 and Table 4. Similar to the results of 3, 4, 5-component circuits, SFM has higher success rates in low tolerance ranges (0.01 to 0.03) compared to FM. This shows that our succinct formulation also enhances low precision sensitivity in transferability evaluation.

Models trained with SFCI demonstrate best transferability. Notably, the success rate of the model finetund with 2000 6-component circuits improves to 0.84 compared to FM's 0.76 under tolerance 0.1. In MSE, SFCI also performs the best compared to all other formulations with up to 58.5% improvement. This result shows that the sparse graph representation is more suitable for complex circuits due to its short token length. Additionally, our component-type tokens let model to better learn different component's representations to boost the generalizability. In summary, these results suggest that our proposed SFCI is the best formulation to perform the topology generation.

*Table 5.* MSEs of voltage conversion ratio and efficiency evaluated on SFCI and its two variants using 3, 4, 5-component circuits: (1) SFCI-NDP and (2) SFCI-NCT, as in Figure 4.

| MSE | Voltage | Eff |
|---|---|---|
| SFCI-NDP | 0.0033 | 0.0028 |
| SFCI-NCT | 0.0012 | 0.0022 |
| SFCI | 0.0006 | 0.0002 |

## 6. Ablation Study & Discussion

### 6.1. SFCI without Duty-Cycle Prefix

When shrinking formulations, the prefix containing five duty cycle numbers is a common feature across data points and might be removed, as stated at the end of Section 4.2. Thus, we evaluate how omitting this prefix affects model performance by training a variant SFCI-NDP without the duty-cycle prefix, as in Figure 4. The results in Figure 8 and Table 5 show that SFCI-NDP suffers a performance drop compared to SFCI, showing that this prefix provides an effective guidance for model's decision making.

### 6.2. SFCI without Component-Type Tokens in Output

To validate the design choice of keeping component-type tokens in the output of SFCI, we compare SFCI against a variant SFCI-NCT that removes them from the output as

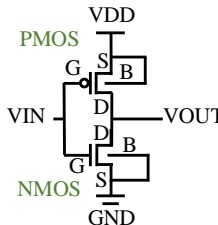

**LM input:**
{Performance specification} <sep> <VDD> <VIN> <GND> <PMOS> 0 S <PMOS> 0 G <PMOS> 0 D <PMOS> 0 B <NMOS> I S <NMOS> I G <NMOS> I D <NMOS> I B <sep>
**LM output:**
{parameters to be set} <sep> <VDD> <PMOS> 0 S <PMOS> 0 B , <VIN> <PMOS> 0 G <NMOS> I G ,<GND> <NMOS> I S <NMOS> I B , <VOUT> <PMOS> 0 D <NMOS> I D <sep>

*Figure 9.* The example of extending SFCI into a transistor-based inverter that contains an NMOS device and a PMOS device, each has four distinct device pins: drain (D), gate (G), source (S), and body (B). We view an NMOS with the identifier `0` as four nodes: `<NMOS> 0 D`, `<NMOS> 0 G`, `<NMOS> 0 S` and `<NMOS> 0 B`. In addition, a PMOS device with the identifier `1` is viewed as four nodes: `<PMOS> 0 D`, `<PMOS> 0 G`, `<PMOS> 0 S` and `<PMOS> 0 B`. Then, we base on all nodes to construct our inverter.

shown in Figure 4. SFCI-NCT yields worse performance according to both Figure 8 and Table 5. These results confirm that including component-type tokens helps the model better capture the circuit structure, leading to more accurate generation despite a slightly longer output.

### 6.3. Computation Efficiency Analysis

We analyze the computational efficiency of our SFCI by comparing it with the SFM. Specifically, the training for SFM saturates at 8,943 steps, whereas SFCI converges significantly faster at 6,886 steps, representing a 23.0% reduction in required training steps. Moreover, as the token length complexity decreases to $O(|V|)$, SFCI also achieves shorter inference times compared to SFM. These results demonstrate that SFCI substantially enhances computational efficiency in both training and inference phases.

### 6.4. Extend SFCI into Transistor-based Circuits

Future developments can extend SFCI to handle transistor-based circuits, enabling support for a wider range of applications. In this context, transistors, e.g., NMOS devices (ND), have four distinct pins: drain (D), gate (G), source (S), and body (B). To represent these circuits, nodes need to be defined at the pin level. Thus, we can view an NMOS with the identifier `0` as four nodes: `<NMOS> 0 D`, `<NMOS> 0 G`, `<NMOS> 0 S`, `<NMOS> 0 B`. We construct an example of transistor-based inverter in the Figure 9.

## 7. Conclusion

In this paper, we introduced LaMAGIC2, an advanced circuit formulation designed to enhance language model-based topology generation for analog circuit design. LaMAGIC2 addresses key limitations of prior approach by introducing the Succinct Float-input Canonical Formulation with Identifiers (SFCI). SFCI improves component-type recognition through the use of unique identifiers, simplifies input specifications to enhance numeric precision sensitivity, and reduces circuit representation length to $O(|V|)$. Experimen-

tal results show that SFCI achieves a 34% higher success rate under a stringent tolerance condition 0.01 compared to LaMAGIC's FM formulation, along with 10X lower MSEs. Moreover, SFCI excels in transferability evaluations, achieving up to 37.5% higher success rates and 58.5% lower MSEs when finetuned on limited datasets of more complex circuits. LaMAGIC2 paves the way for more efficient and scalable automated analog design methodologies.

For future works, first, we can integrate search-based decoding methods with our models to generate optimized circuits for even larger and more diverse design spaces. This integration can balance the strengths of generative and search-based approaches, enhancing the quality and practicality of automated analog circuit design solutions. Second, we can extend our formulations to transistor-based circuits to support wide range semiconductor circuit design applications that has precise numeric specifications from users. LaMAGIC2 presents great potential for complex circuit designs.

## Acknowledgment

This work is partially supported by SRC 3104.001 and NSF 2106828. Special thanks to Prof. Jason Cong from UCLA for paper writing suggestion.

## Impact Statement

This paper presents work whose goal is to advance the field of Machine Learning. There are many potential societal consequences of our work, none of which we feel must be specifically highlighted here.

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
