# OpenReview forum: "LaMAGIC2: Advanced Circuit Formulations for Language Model-Based Analog Topology Generation"
_ICML.cc/2025/Conference — ICML 2025 poster_

### Official Review · Reviewer_F8r4 · 2025-03-11

**Overall Recommendation:** 3

**Summary:**

This paper introduces LaMAGIC2, a circuit formulation approach analog topology generation. The authors identify limitations in previous methods, particularly LaMAGIC, which used inefficient circuit representations with quadratic token length complexity and showed low sensitivity to numeric input precision. Experimental results show that LaMAGIC2 achieves 34% higher success rates under tight tolerance conditions (0.01) and 10X lower MSEs compared to LaMAGIC.

**Claims And Evidence:**

Claim: LaMAGIC2 achieves higher success rates under tight tolerance conditions.
Evidence: The authors provide comprehensive experiments comparing success rates across different tolerance levels (0.01-0.1), showing LaMAGIC2 outperforming baselines.

Claim: SFCI reduces token length complexity to O(|V| + |E|).
Evidence: The authors provide theoretical analysis and empirical measurements of token lengths (Table 1), showing significant reduction compared to matrix-based formulations.

Claim: Component-type tokens improve circuit structure learning.
Evidence: The ablation study comparing SFCI with SFCI-NCT (without component-type tokens) provides clear evidence supporting this claim.

**Essential References Not Discussed:**

Recent work on graph generation using language models beyond circuit design, which could provide additional context for the token-based graph representation approach.

**Experimental Designs Or Analyses:**

The experimental design is generally sound: The authors use the same dataset as LaMAGIC for fair comparison, with appropriate splits for training and evaluation. Model architecture: Using the same Flan-T5-base architecture as LaMAGIC ensures fair comparison. The simulation-based evaluation using NGSPICE provides realistic performance metrics. The approach of training on 3,4,5-component circuits and fine-tuning on limited 6-component circuits is a reasonable test of transferability.

**Methods And Evaluation Criteria:**

- Circuit formulations: The authors provide clear descriptions and analyses of different formulations, highlighting their advantages and disadvantages.
- Comparison with baselines: The authors compare with both previous formulations from LaMAGIC and an RL-search method, providing a comprehensive evaluation.
- Transferability evaluation: Testing on more complex 6-component circuits with limited training data is a practical approach to evaluate generalization capabilities.

**Other Comments Or Suggestions:**

NA

**Other Strengths And Weaknesses:**

Strengths:

The paper presents a clear analysis of the limitations of existing formulations before proposing improvements.
The proposed SFCI formulation is elegant and addresses multiple issues simultaneously.
The experimental results are comprehensive and convincingly demonstrate the advantages of the proposed methods.

Weaknesses:

The paper could benefit from more discussion of failure cases or limitations of the proposed formulations.
While the paper shows improved performance on 6-component circuits, it's unclear how well the approach would scale to even larger circuits.

**Questions For Authors:**

Have you investigated how the choice of language model architecture influences the performance of various formulations? For instance, do models with longer context windows alter the relative benefits of SFCI compared to matrix-based formulations?

**Relation To Broader Scientific Literature:**

It builds upon prior work in search-based approaches for analog topology design.

**Theoretical Claims:**

The paper does not present formal mathematical proofs but makes theoretical claims about computational complexity.

---

> ### Author Rebuttal · Authors · 2025-03-31
>
> ### Addressing limitations
> Thanks for your advice on discussing the weakness of our methods. This is a helpful idea to contribute more to the community.
>
>
> The circuit space will grow exponentially with number of nodes increases. In addition, the simulation time for larger circuits will also increase, causing the scarcity of training data and limiting the model generalizability. Thus, this one-shot generation approach may become less effective for significantly larger circuit designs.
>
>
> To address these challenges, for future works, we are developing search-based decoding methods with our models to generate optimized circuits for larger and more diverse design spaces. For example, Monte-Carlo tree search (MCTS) is a promising method to expand the language model capability in test-time computation. This integration can balance the strengths of generative and search-based approaches, enhancing the quality and practicality of automated analog circuit design solutions.
>
> ### Response to Question on Architecture Impact
>
> In our experiments, all circuit representations (both SFCI and matrix-based) fit within a single 1024-token context window, and no context-window shifting is necessary during generation. Under this setting, we observed that our SFCI consistently outperformed matrix-based formulations. The reduced token length and the use of component-type token of SFCI can improve the model performance and has faster convergence (23.0% fewer training steps compared to SFM, as mentioned in our response to Reviewer 2sgz).
>
>
> Given that our experimental setup does not require multiple context windows, increasing the window size (e.g., from 1024 to 2048 tokens) would not directly influence the relative benefits of SFCI. However, we expect that as circuit size scales up (e.g., larger circuits exceeding one context window), SFCI would provide greater advantages. Specifically, when matrix-based formulations need context-window shifting, this will lead to degraded performance, since the model cannot see the entire circuit during generation. In contrast, SFCI would be more suitable for generating larger circuits without context-window shifting due to its compact token length.

---

### Official Review · Reviewer_2sgz · 2025-03-12

**Overall Recommendation:** 4

**Summary:**

This paper introduces LaMAGIC2, an improved approach for language model-based analog topology generation. It proposes SFCI, which enhances component recognition, reduces token complexity from $O(|V|^2)$ to $O(|V| + |E|)$, and improves numerical precision sensitivity. The method achieves a 34% higher success rate under strict tolerance and 10X lower MSE compared to prior work. It also generalizes better to complex circuits, improving MSE by up to 58.5%. Future work includes extending SFCI to transistor-based circuits and integrating search-based techniques.

**Claims And Evidence:**

The paper claims that LaMAGIC2 reduces token complexity, improves component recognition, enhances numerical precision, and increases transferability through a sparse representation. These claims are well-supported, as the method effectively streamlines circuit encoding, improves model learning, and aligns with real-world circuit sparsity.

**Essential References Not Discussed:**

As far as I am concerned, the references are sufficient.

**Experimental Designs Or Analyses:**

The experimental design is generally sound, with well-defined metrics, meaningful comparisons, and ablation studies. Success rate and MSE effectively assess circuit generation, and the LaMAGIC dataset provides a reasonable benchmark. Comparisons with prior methods, including RL-based search and different formulations, demonstrate performance gains, while ablation studies on SFCI variants validate key design choices.

**Methods And Evaluation Criteria:**

The proposed methods and evaluation criteria align well with language model-based analog topology generation. SFCI effectively reduces token complexity, improves component recognition, and enhances numerical precision, all essential for accurate circuit generation. Success rate and MSE metrics appropriately measure performance, while the benchmark dataset from LaMAGIC provides a diverse set of 3–6 component circuits for evaluation.

**Other Comments Or Suggestions:**

The claim that reducing token complexity from $O(|V|^2)$ to $O(|V| + |E|)$ improves efficiency is intuitive, as shorter sequences generally lead to faster inference and reduced memory usage in transformer models. However, the paper lacks explicit empirical analysis to substantiate this impact. A more detailed discussion on how token reduction influences training time, convergence speed, and inference efficiency would strengthen the argument.

**Other Strengths And Weaknesses:**

The paper is clearly written, effectively identifying the limitations of previous methods and explaining how each issue is addressed. The structured presentation makes it easy to follow the improvements and their impact on circuit generation.

Weaknesses are shown in other parts.

**Questions For Authors:**

Can you provide some results in terms of computational efficiency, e.g., training time or convergence speed compared to previous methods?

**Relation To Broader Scientific Literature:**

The paper builds on LaMAGIC and related work in language model-based circuit generation, improving token efficiency, numerical precision, and transferability by incorporating structured tokenization, float-input formulations, and efficient graph encoding, aligning with broader trends in transformer-based design automation.

**Theoretical Claims:**

The paper does not present formal proofs for its theoretical claims but instead supports its methodology through empirical results. The key theoretical claim is that reducing token complexity from $O(|V|^2)$ to $O(|V| + |E|)$ improves efficiency and model performance. While this claim is reasonable given the sparsity of real-world circuits, it is not formally proven. The paper demonstrates the impact of this reduction through token length statistics (Table 1) and empirical success rate/MSE comparisons (Tables 2–5), but it does not provide a mathematical proof showing how this complexity reduction affects training dynamics or inference efficiency.

---

> ### Author Rebuttal · Authors · 2025-03-31
>
> ## Question for computational efficiency
>
> Based on the question, we further record the training steps that required for matrix formulation (SFM) and the succinct canonical formulation with identifier (SFCI). Specifically, SFM saturates at 8943 steps, and SFCI converges at 6886 steps. This shows that our SFCI reduces 23.0% of training steps compared to SFM, thus enhancing the computational efficiency.
>
>
> We will include this experimental results if we get accepted!

---

> > ### Comment · Reviewer_2sgz · 2025-04-02
> >
> > Thank you for your reply, which addressed my concerns. I will retain the score.

---

### Official Review · Reviewer_B3t1 · 2025-03-14

**Overall Recommendation:** 2

**Summary:**

This paper addresses the automation of analog topology design, which aims to determine the optimal connections between given nodes while satisfying various constraints. Existing methods, including search-based and reinforcement learning-based approaches, are often inefficient. This paper analyzes the structure of a large language model-based approach, LaMAGIC. To tackle the challenges of low-precision sensitivity and long token lengths, the paper introduces two circuite representations:SFM and SFCI. SFM simplifies numerical input representation, while SFCI preserves a sparse graph structure with improved component-type recognition, leading to more efficient and accurate circuit generation. The proposed method, LaMAGIC2, outperforms baseline approaches by achieving higher success rates under strict tolerances, reducing mean squared errors by up to 10×, and demonstrating superior transferability to more complex circuits, with improvements of up to 58.5%.

**Claims And Evidence:**

The proposed methods, SFM and SFCI, are well supported by Figure 3, which also effectively illustrates the disadvantages of previous methods.

**Essential References Not Discussed:**

All essential references are covered; however, additional related works should be included to provide a more comprehensive background and context for the study.

**Experimental Designs Or Analyses:**

The author adhered to the experimental design of previous studies, ensuring the validity of the results. Additionally, while the ablation studies are appropriately conducted, it would be necessary to perform experiments using a different LLM model architecture, given that this paper is primarily experimental.

**Methods And Evaluation Criteria:**

The proposed method is reasonable, but it primarily involves prompt engineering, which is a relatively naive mechanism. The approach presented by the author is an extension of the previous method, but a more advanced algorithm is needed for further improvement. Nevertheless, the evaluation criteria are appropriately set.

**Other Comments Or Suggestions:**

There are no further comments for the author.

**Other Strengths And Weaknesses:**

This paper presents an effective method that integrates large language models (LLMs) to automate analog topology design. The proposed approach is practical and has a significant impact on automation system design. However, the method is somewhat naive, as it primarily relies on prompt engineering.

**Questions For Authors:**

1. Why the author specify 'LaMAGIC' for the baseline algorithm? Is there no other similar research to handle the limitation of the LaMAGIC?
2. Can the author give a more detailed explanation of how the author changed the tocken length complexity to O(|V|+E|)? It is quite confusing.

**Relation To Broader Scientific Literature:**

The author discusses search-based and RL-based methods; however, to provide a comprehensive understanding of the automation of analog topology design, more related work should be included. Additionally, sufficient information on analog topology design using LLMs is needed. The author refers to "one such approach" for the direct mapping from performance requirements to circuit topologies, but a brief explanation of why LaMAGIC was chosen for this study would further clarify the rationale behind the selected method.

**Theoretical Claims:**

There is no theoretical validation required, but it would be beneficial to analyze or establish a connection to the LLM structure to explain why the proposed method achieves a higher success rate. Additionally, providing a reference that supports the main idea would further strengthen the argument.

---

> ### Author Rebuttal · Authors · 2025-03-31
>
> ## Clarification on previous work and our methodology
> We thank the reviewer for this important comment. Our paper focuses on developing supervised fine-tuning (SFT) methods for language models in analog topology generation task. Our SFCI formulation contributes these key innovations:
> 1. It proposes a compact canonical form with component-type tokens to enhance the learning of different circuit devices.
> 2. It increases the attention on the float-valued inputs of circuit specifications, thus effectively capturing the relations between numerical circuit specifications and circuit topologies.
>
>
> LaMAGIC was selected as the primary baseline because, to the best of our knowledge at submission time, it is the only prior work using SFT with transformer-based LMs for analog topology generation. Additionally, we compare with a RL search method for power converter to comprehensively demonstrate the benefits of our generative approach over conventional search techniques. These two methods (LaMAGIC and RL) cover well the previous works for power converter topology generation, and we have thoroughly discussed other related methods in Section 2.3 of our paper.
>
> ## Question for complexity
> We appreciate the reviewer’s question and apologize for any confusion caused by our initial complexity characterization. In our original submission, we described the token length complexity as $O(|V| + |E|)$, assuming a simple graph scenario where the SFCI representation is a vertex-to-edge adjacency list for a graph with $|V|$ vertices and $|E|$ edges.
> Upon careful re-examination, we realized this characterization was inaccurate, as our formulation is actually a hypergraph represented by an edge-to-vertex adjacency list.
>
> ### Correct Hypergraph Complexity Analysis
> Each hyperedge $e_i$ has $k_i$ vertices. Therefore, the total token length, representing the sum of vertex incidences across all hyperedges, is:
> $$
> \sum_{i=1}^{|E|} k_i
> $$
> For our power converter devices, each vertex appears in at most two hyperedges: $d(v)≤2$.
>
>
> Thus, the total number of vertex incidences , which equals to the number of edges of all nodes, is:
> $$
> \sum_{i=1}^{|E|} k_i = \sum_{v \in V} d(v) \leq 2|V|
> $$
> This upper bound implies the total token length is at most $2|V|$, which simplifies to $O(|V|)$. This result indicates that the token length complexity of SFCI is independent of the number of edges $|E|$.
> The $O(|V)$ token length of SFCI demonstrates its compactness compared to $O(|V|^2)$ lengths of matrix formulations.
>
> We will revise the token length complexity to $O(|V|)$ and include the above proof in our manuscript accordingly, to improve clarity and readability for the readers.

---

### Official Review · Reviewer_bUno · 2025-03-15

**Overall Recommendation:** 3

**Summary:**

This paper proposes LaMAGIC2, introducing succinct formulations (SFM and SFCI) for language-model-based analog circuit topology generation. Compared to the previous LaMAGIC approach, these formulations effectively reduce output sequence length and improve component-type recognition. Experiments demonstrate that LaMAGIC2 achieves higher success rates and lower MSE under strict tolerance conditions.

**Claims And Evidence:**

The claims made in the submission are well-supported by clear and convincing evidence.

**Essential References Not Discussed:**

The paper provides a sufficient discussion of prior work, particularly LaMAGIC, and appropriately cites relevant literature. No essential references appear to be missing.

**Experimental Designs Or Analyses:**

The experimental design is sound, using benchmark datasets and appropriate metrics.

**Methods And Evaluation Criteria:**

The methods and evaluation criteria are well-aligned with the problem, effectively measuring improvements in circuit topology generation.

**Other Comments Or Suggestions:**

1. The paper focuses primarily on power converter circuits. Expanding experiments to other types of analog integrated circuits or evaluating the method’s transferability to new circuit types would strengthen its generalizability.

2. The proposed approach is tested on circuits with up to six components, which is only a slight scalability improvement over previous work (five components). Evaluating larger-scale circuits would better demonstrate the method’s scalability.

**Other Strengths And Weaknesses:**

The paper presents a well-structured study with clear contributions to refining analog circuit topology generation using language models. The proposed formulations effectively improve efficiency and component-type recognition. While the methodological novelty is somewhat incremental, the improvements in output sequence length and numerical precision make it a valuable contribution.

**Questions For Authors:**

1. The paper uses T5 as the underlying model. Have the authors tried or considered extending their formulations to other language models (e.g., GPT variants)? How generalizable is the proposed approach across different model architectures?

2. Have the authors conducted comparative experiments with advanced commercial LLMs (e.g., GPT-o1, Claude 3.7, etc.) in circuit topology generation tasks?

**Relation To Broader Scientific Literature:**

The paper builds on prior work in language-model-based analog circuit topology generation, particularly LaMAGIC. It improves upon previous formulations by reducing output sequence length and enhancing component-type recognition. These contributions align with broader trends in applying machine learning, especially transformer-based models, to circuit design automation. However, expanding comparisons to other ML-based circuit design approaches could further contextualize its impact.

**Theoretical Claims:**

No theoretical claims were involved.

---

> ### Author Rebuttal · Authors · 2025-03-31
>
> ### Q1: Choice of Model Architecture
> We appreciate the suggestion to evaluate our proposed formulations using other model architectures. In this work, we adopted T5 to maintain architectural consistency with previous work LaMAGIC, enabling a fair comparison focused on the impact of new formulations.
>
> We have considered extending to encoder-only models (e.g., GPT-2), which are often more suitable for scaling up the parameter count compared to encoder-decoder models. However, due to the limited rebuttal timeline, we were unable to complete the experiments.
>
>
> Our proposed float-input setting and SFCI formulation are architecture-agnostic and can be applied by any transformer models with sequential inputs. Specifically, the float-input directly feeds numerical specifications into the model without discretization by transitional tokenizers to improve numerical representation. Also, the SFCI formulation further provides a compact representation of circuit topologies, which improves the learning for larger circuits.
>
> Another reason we chose T5 is its smaller model size comparing to more general models, which makes it more suitable for application-specific use cases like this work while also being more energy-efficient and low-cost.
>
> ### Q2: Comparison with commercial LLMs
>
> Based on your suggestion, we evaluate o1 using a few-shot setup. We provide 100 example circuits as context and evaluate performance on the same 350 specifications used in our main experiments (Figure 5 in the paper). Below are the results:
>
> Success rates at thresholds from 0.01 to 0.1:
>
> |Threshold| 0.01| 0.02 | 0.03 | 0.04 | 0.05 | 0.06 | 0.07 | 0.08 | 0.09 | 0.1|
> | -------- | ------- | ------- | ------- | ------- | ------- | ------- | ------- | ------- | ------- | ------- |
> |Success rate of o1 | 0.27 | 0.32 | 0.36 |  0.39 |  0.40 |  0.41 |  0.42 |  0.43 |  0.44 | 0.46 |
> |Success rate of SFCI | 0.90 | 0.96 | 0.98 | 0.99 | 0.99 | 1.00 | 1.00 | 1.00 | 1.00 | 1.00 |
>
> MSE of voltage conversion ratio:
> | Metric | SFCI | o1 |
> |:------:|:-------------:|:--:|
> |  MSE   |     8e-5      | 0.602 |
>
> This result shows that our model significantly outperforms o1, indicating our supervised fine-tuning is needed to tackle the analog circuit design, which demand high precision and the interaction of custom SPICE simulators.

---

### Decision · Program_Chairs · 2025-05-01

**Decision:**

Accept (poster)

**Comment:**

This paper proposes a novel method for large-language-model-based automatic circuit analog topology design. The method builds on a previous method, LaMAGIC, and proposes nontrivial technical improvements including supervised finetuning, compact and better token representations for the output, etc. Issues regarding contribution, evaluation, and lack of details were raised by reviewers, but were addressed satisfactorily. The complexity claim was questioned by several reviewers, but was revised by authors during rebuttal. Empirical results were solid and strong, as agreed by all reviewers.